# Procedure to Obtain Cu_2_O Isolate Films, Structural, Electrical, and Morphological Characterization, and Its Use as an Electrical Isolator to Build a New Tube Furnace

**DOI:** 10.3390/ma16041361

**Published:** 2023-02-06

**Authors:** Hernando Correa, Ricardo Pineda Sánchez, Diego Peña Lara

**Affiliations:** 1Instituto Interdisciplinario de las Ciencias, Universidad del Quindío, Armenia 630 004, Colombia; 2Grupo de Transiciones de Fases y Materiales Funcionales, Departamento de Física, Santiago de Cali 760 032, Colombia; 3Centro de Excelencia en Nuevos Materiales (CENM), Universidad del Valle, Santiago de Cali 760 032, Colombia

**Keywords:** electrical resistance furnace, Cu2O, electrical insulation film, thermal oxidation, copper applications, electric resistance heating

## Abstract

Copper oxide is a widely studied compound in wastewater decontamination, hydrogen production, solar cell production, and sensor fabrication. In recent years, many architectures and structures with the potential for developing clean technologies have been synthesized. A procedure by thermal oxidation to grow electrical insolate Cu${}_{2}$O films on copper surfaces in an air atmosphere was developed. The results of the morphological and structural characterization of the copper oxide layers evidence the presence of Cu${}_{2}$O polycrystalline films. The films have polyhedral architectures of approximately 1.4 μm thickness and are electrically insulating. A novel copper resistive furnace was built using this copper oxide film which was used as an electrical insulator between the electrical resistance of the heater and the surface of the copper thermal block. The application improves the efficiency of the resistive furnace in terms of the temperature reached and the thermal coupling response time relative to the performance of conventional furnaces using ceramic insulation. Over the entire operating temperature range explored for the same power supply, the copper oxide-coated furnace achieved higher temperatures and faster response times than the traditionally coated furnace.

## 1. Introduction

Copper is a metal whose mechanical, thermal, and electrical properties, recyclability, and worldwide availability provide several advantages for applications in the construction of electrical networks, anticorrosion films, and heat exchangers in renewable energies [1]. Its electrical and thermal properties and those of its native oxides have been extensively studied. The electrical conductivity (σ) of copper oxide has been measured in various temperatures and oxygen pressure ranges and for different crystallographic forms [2,3]. The stoichiometric deviation from cuprous to cupric oxide was studied by thermogravimetric analysis technique [4], the chemical analysis of quenched samples [5] as a function of temperature and oxygen partial pressure, among others.

Copper oxidation was extensively studied for several decades, with work reported as early as the 1920s [6]. Two families of stable copper oxide phases are recognized: cuprous oxide (Cu${}_{2}$O) and cupric oxide (CuO). By the thermal oxidation method, on copper surfaces, in an air atmosphere or oxygen, the formation of multiple Cu${}_{2}$O, CuO, and Cu${}_{3}$O${}_{2}$ phases is observed, depending on the thermodynamic stability of the oxides, the purity of the reagents, the atmosphere, and the thermal history [2,7,8,9,10,11]. Among industrial applications [7,12,13,14,15,16,17,18,19,20,21] are the manufacture of transparent conductive layers (electrodes), the construction of semiconductor devices (gas sensors), the design of electronic devices (heterostructures), multilayer structures (fuel cells), supercapacitors, batteries, catalysis, and CO${}_{2}$ reduction processes.

Cu${}_{2}$O is reported as a p-type semiconductor with a cubic structure and a gap of 2.1 eV [22,23,24], with electrical properties controlled by an acceptor level above the valence band (energy of 0.4 eV) and two donor levels near the conduction band at −1.1 and −1.3 eV. In recent years, it was reported that Cu${}_{2}$O presents different crystalline architectures [7] and electrical properties [3,25] that depend on its architecture. CuO is a semiconductor with a monoclinic structure 1.2 eV gap and can be synthesized in different nanostructures: nanowires, nanowires, and nano-bigots, among others, have been reported. Both oxides are p-type semiconducting materials with cation vacancies and electron-hole primary defects. Another compound formed by copper and oxygen is Cu${}_{3}$O${}_{2}$, an oxide reported as a metastable phase of Cu${}_{2}$O [26,27,28,29,30].

Heat treatments on Cu${}_{2}$O produce the formation of crystalline films, with micrometer-sized crystals presenting facet architectures such as those reported in Ref. [7]. In this work, it was observed that the electrical conductivity depends on the facet architecture. Tan et al. [3] studied the electrical characteristics of copper oxides showing that the octahedral crystal is highly conductive, the cubic crystal is moderately conductive, and the rhombic dodecahedron crystal is a non-conductor. The differences in conductivity are attributed to the electronic band structures; changing the crystallographic structure changes the curvature of the bands and the height of the potential barriers at the interfaces [3]. Roy and Wright [31] reported that the electrical conductivity in Cu samples decreases upon oxidation; therefore, the resistance at room temperature after oxidation is much higher, evidencing a slow transition from metal. De los Santos Valladares et al. [29] reported the electrical characteristics of Cu${}_{2}$O (cubic structure) and CuO (monoclinic structure) thin films obtained by thermal oxidation, evidencing the increase in the electrical resistance of the system upon oxidation.

In different technological applications, it is necessary to have electrically isolating copper surfaces. For this purpose, a wide variety of electrical insulators are available, such as polymers, paints, and enamels. There is a variety of copper oxides, but these oxides are usually semiconductors and cannot be used as electrical insulators [3,25,29,30,32,33,34].

In this work, we report a thermal oxidation procedure to produce an electrically insulating Cu${}_{2}$O film grown on a copper surface. The electrical, structural, and morphological properties were characterized using electrical impedance, XRD, and SEM techniques. We use the Cu${}_{2}$O electrical insulating film to build a novel prototype of a resistive furnace (CuO${}_{2}$ insulation tube furnace, CuITF) that uses this film as insulation between the copper thermal block interface of the furnace and the electrical heating element. Thus, it guarantees a better thermal coupling and a more efficient thermal response of the CuITF compared to a conventional or ceramic cover insulation tube furnace (CITF).

## 2. Materials and Methods

Cylindrical commercial copper (ASTM B 280, 99.5% purity) was used to manufacture the furnace thermal blocks. The chemical analysis of the samples on an Armstrong V950 spark spectrometer showed that the material contained small amounts of Se, Zn, Pb, Fe, Sn, Ni, Al, Sb, As, Be, Bi, Mn, P, Si, Co, and Cr. Therefore, the purity of copper is 99.50% and corresponds to a sample classified as copper 2N5 (impurities of Si, Sb, and Zn have the highest percentage). The 4.76 mm external diameter oxidized cylindrical samples which were cut into sections between 3 and 10 cm in length, whose surfaces were polished with abrasives until a specular surface was achieved and then washed with a cleaning solution in an ultrasound chamber. Copper oxide films were obtained on the surface by a thermal oxidation procedure, subjecting the samples to temperature runs in a muffle under an air atmosphere, controlling the exposure time, temperature change rates, and final temperature. The samples were subjected to two thermal stages: first, the copper tube is subjected to thermal pre-oxidation by heating in a muffle at room temperature to reach the pre-oxidation temperature (between 260 and 280 ${}^{{}^{\circ}}$C), at a heating rate between 10 and 15 ${}^{{}^{\circ}}$C/min, and second, from this temperature, at a rate between 1 ${}^{{}^{\circ}}$C/min and 3 ${}^{{}^{\circ}}$C/min, until the oxidation temperature reached between 330 and 340 ${}^{{}^{\circ}}$C. The samples remained at the oxidation temperature for between 45 min and 55 min. Once the oxidation procedure was completed, the heat treatment was stopped so that the formation of undesired phases would not continue, obtaining the required phase (electrically insulating Cu${}_{2}$O), cooling the samples to 50 ${}^{{}^{\circ}}$C at a rate between 1 and 3 ${}^{{}^{\circ}}$C/min. The films obtained have a 1.4±0.1 μm thickness.

Scanning beam diffractograms on a PANalytical multi-purpose diffractometer model X’Pert PRO MRD and a Bruker D8 Advance X-ray to identify the oxides present in the electrically insulating copper film diffractometer (coupled mode) were performed. The morphological properties of the oxidized copper surfaces were measured by scanning electron microscopy (SEM) on an FEI QUANTA 250 equipment. Electrical impedance measurements of the copper oxide samples were performed on a HIOKI 3532-50 impedance meter at room temperature under an air atmosphere with a frequency range between 1 Hz and 5 MHz. Copper tape electrodes were used; the tape was rolled transversely over the copper tubes, the tape being in electrical contact with the oxide film.

The surface of a copper cylinder was oxidized to obtain a copper oxide film that acts as an insulating layer, avoiding the use of ceramic materials as electrical insulation. The thickness of this insulating layer is approximately one micron, in contrast to other refractory insulations with thicknesses of the order of millimeters. By using an oxidized cylinder, the CuITF was made. The CuITF heater has a ferronickel wire resistor (Kanthal A1) wound over the oxidized copper surface and is coated with a Ceramvest ceramic layer to prevent further oxidation. This application substantially improves the efficiency.

## 3. Results and Discussion

Figure 1a displays the oxidized copper tube of 4.76 mm external diameter and 80 mm long, and Figure 1b shows another 15 mm internal diameter, 21 mm external diameter, and 100 mm long tube. Both were oxidized by the thermal oxidation method, obtaining the reproducibility of the oxide film and preserving its characteristics of adhesion and electrical insulation over time.

Figure 2 shows the XRD diffractograms of copper (black line) and three oxidized films under the same experimental conditions, identified as S1 (sample 1, green line), S2 (sample 2, red line), and S3 (sample 3, blue line). The observed peaks correspond to Cu (orange squared) and Cu${}_{2}$O (dark gray circle). In all samples, there are the characteristic peaks of metallic copper as a substrate material and the diffraction maxima of cuprous oxide (PDF 01-078-2076). A low-intensity peak at 25.3${}^{{}^{\circ}}$, as indicated by the vertical arrow, identifies the Cu${}_{3}$O${}_{2}$ [35] phase or some substrate impurity such as CuHO${}_{2}$ or Cu-P-O. D.E. Mencer et al. [27] recommended performing electrochemical measurements such as linear scanning voltammetry (LSV) to identify other phases such as Cu${}_{3}$O${}_{2}$ [10,26].

Figure 3 exhibits the morphology of the insulating film obtained by SEM; observe the formation of Cu${}_{2}$O crystals with different facets. The films have polyhedral architectures. The film thickness is between 1.355 and 1.442 μm.

The electrical impedance measurement of an insulating Cu${}_{2}$O film in Figure 4 is presented. The fitting circuit model using a parallel circuit shows that the film has a DC resistance of order $9\times 10^{3}$ MΩ (red curve) which corresponds to the resistance of an insulating material. The inset compares the insulating oxide film with a semiconducting Cu${}_{2}$O film which presents a resistance of 100 KΩ (black curve), corroborating that the Cu${}_{2}$O film obtained by thermal synthesis has an electrical insulating behavior. The literature shows evidence of a relationship between facet architecture and the electrical conductivity properties of Cu${}_{2}$O crystals [3]. In our insulating films, we observed a mixture of polyhedral architectures.

### 3.1. Technical Comparison between the CuITF and CITF Prototypes

Figure 5 shows the steps to make the two prototypes of furnaces (CuITF and CITF), which are similar in all aspects except for the insulation separating the copper surface from the electrical resistance. The left column illustrates the fabrication of the CuITF and the CITF on the right. The unoxidized copper tube precursor material is illustrated in Figure 5a.

Figure 5b shows the copper tube covered with an insulating copper oxide layer. The CITF prototype was coated with a thin ceramic layer (0.12±0.03 mm thickness) as shown Figure 5c, ensuring good electrical insulation and thermal coupling. The resistance of the wire used in Figure 5d,e as an electric heater for the fabrication of the two furnaces was 22 Ω. Figure 5f,g show the CITF and CuITF, which is covered with the same amount of ceramic. The ceramic placed on the CuITF surface prevents additional oxidation when the prototype is heated. The size of the copper tube used in constructing the two furnaces was the same, obtained from the same commercial copper sample complying with the ASTM B 280 standard. For the technical characterization of each furnace, the temperature reached by the furnaces, the power dissipated by the heating resistance, and the time constants, delay time, rise time, settling time, dead time, and gain were calculated. Open-loop heating tests were performed for the furnaces to verify the performance. The temperature was measured inside the cylindrical copper tubes (central region) using a 340 lakeshore temperature controller. The supply voltage of the resistance of the furnaces was 9.6 V. The tests were performed in a desiccant hood under an air atmosphere. The voltage and current data supplied to the furnaces (measured with two Keithley multimeters), with software developed in LabView, were acquired.

Figure 6 shows the reached temperature curves as a function of energy consumed for each furnace. The curves show that, with a similar electrical energy supply for both furnaces, the CuITF reaches an average temperature 13.6% higher than the CITF for a certain representative energy value of 5 kJ in the range of temperature stabilization.

Figure 7a shows the temperature changes in the CuITF as a function of time. The inset shows the 4.0 s delay (dead time $\tau_{d}$) from when the furnace is supplied with 4.1 W of power until the furnace starts to respond. Figure 7b shows a $\tau_{d}=6.3$ s for the CITF at the same power. The shorter time for the CuITF is due to the use of copper oxide as an insulator.

### 3.2. Heating Curve Settings

The heating dynamic of furnaces exhibits behavior characteristic of first-order systems and is fitted by the following equation:(1)$$T\left(t\right)=T_{f}\left(1-e^{-t/\tau }\right)$$
where *T* is the temperature, $T_{f}$ is the final temperature, and *t* is the heating time. The time that shows the velocity of a system at a given input to reach a steady state or when the system reaches 63.2% of the final value (steady state) is called the time constant τ, which is related as:(2)$$\tau =R_{T}C_{T}$$
with $R_{T}\left(\equiv \mathrm{change}\;\mathrm{in}\;\mathrm{temperature}\;\mathrm{difference}/\mathrm{change}\;\mathrm{in}\;\mathrm{heat}\;\mathrm{flow}\;\mathrm{rate}\right)$ being the thermal resistance for heat transfer and $C_{T}\left(\equiv \mathrm{change}\;\mathrm{in}\;\mathrm{heat}\;\mathrm{stored}/\mathrm{change}\;\mathrm{in}\;\mathrm{temperature}\right)$ being the thermal capacitance for conduction or convection heat transfer [36].

Figure 8 shows the heating curves of the CuITF (red line) and the CITF (black line), showing that the CuITF achieved a higher temperature gain for equal supply voltages and currents as the CITF. With the same power supply, the CuITF reaches a temperature of approximately 153.4 ${}^{{}^{\circ}}$C while the CITF is 136.7 ${}^{{}^{\circ}}$C. The fitting for the heating curves of the CuITF and CITF using Equation (Equation 1) shows final temperatures of 151.9 ${}^{{}^{\circ}}$C and 134.2 ${}^{{}^{\circ}}$C, corresponding to a 12% increase in the temperature reached. The delay time was 104.6 s for the CuITF and 125.1 s for the CITF, showing that CuITF is 16.3% faster at reaching 50% of the final temperature than CITF. Figure 8 shows settling times of 636.8 s for the CuITF and 757.2 for the CITF. The obtained gains, calculated as the ratio between the final temperature to the input voltage (9.6 V), were 16.0 and 14.2 for CuITF and CITF, respectively, being 15% higher for CuITF than CITF.

Table 1 shows the fitting modeling parameters used for each furnace.

The CuITF has a lower τ than the CIFT; in other words, its transient response is more optimal. The CuITF heating curve fits very well with the first-order systems model compared to the CITF heating curve (see $R^{2}$ in Table 1).

Table 2 shows other relevant furnace parameters associated with the transient response [36] of the heating curves (see Figure 7a,b and Figure 8, and Table 1).

Figure 9a,b show the power and electrical resistance variations as a function of time for the heating curves shown in Figure 8, respectively. Significant variations in electrical resistance and power supply did not occur during heating. The electrical power dissipated by the CuITF is 2.43% lower than the CITF. The electrical resistance does not vary over time in CuITF, indicating that the copper surface isolation and the oxidized surface do not show changes when the temperature varies. Note the high stability of electrical power dissipation and electrical resistance as a function of time for CuITF.

### 3.3. General Characteristics of the Furnaces

To verify the stability and reproducibility of the heating curves, Figure 10 shows the temperature behavior as a function of time for six runs for two furnaces with 11 Ω electrical resistances under the same experimental conditions and carried out in an air atmosphere. The CuITF presents a higher temperature gain than the CITF. The average temperature for the CuITF is 105 ${}^{{}^{\circ}}$C, and the CITF is approximately 92 ${}^{{}^{\circ}}$C. Both furnaces have a reproducible behavior associated with the stability of both the ceramic layer and the oxide layer, as will be further described subsequently (in the subsection Time Performance Tests). The furnaces fed with 7.1 W power show that the CuITF reaches a temperature 14% higher than the CITF.

### 3.4. Time Performance Tests

Figure 11 shows the reproducibility and durability of CuITF in time over a 2-month interval. The curve shift is due to the feeding start times being different. The final temperature is higher because the furnace was fed with a higher power, verifying that it withstands relatively high temperatures after two months of manufacture.

Building another furnace with 25 Ω electrical resistance under the same conditions as the CuITF, the functional stability of the furnace was established. The furnace underwent experimental tests varying the voltage and power current. The furnace reached different temperatures according to the supplied voltages and currents, observing reproducibility in the heating curves generated under the same experimental conditions. Figure 12a shows eight heating curves as a function of time. Note the reproducibility of the heating curves in runs 6, 7, and 8, where voltages and currents were constant (18.8 V and 0.75 A) up to 2500 s. The furnace reached a temperature of 330 ${}^{{}^{\circ}}$C.

Figure 12b shows the variation of electrical resistance as a function of time. For each heating, electrical resistance fluctuations (25±1.3Ω) show a resistance that does not undergo considerable variations with different heating, voltages, and supply currents, which is evidence of the effectiveness of the oxide film as an electrical insulator. The fact that the resistance does not change after successive heating–cooling runs implies that the oxide coating material does not change its physical properties related to its quality as an insulator.

## 4. Conclusions

We reported a novel procedure for the synthesis by the thermal treatment of a Cu${}_{2}$O electrical insulator film on copper surfaces using appropriate parameters such as heating profiles and oxidation temperature in the air atmosphere. This film has an electrical resistance of the order of the mega-ohm, which is suitable to be used as electrical insulation in the manufacture of furnaces.

The results of the structural characterization by X-ray diffraction of the oxidized copper films show the presence of cuprous oxide (Cu${}_{2}$O). The procedure to obtain the insulating copper oxide film does not require sophisticated technological equipment since the process is carried out in an atmosphere of air and heating at relatively low temperatures. It is an economical and easy-to-implement technique.

SEM measurements showed that the grown films have thicknesses of approximately 1.4 μm, and the SEM images showed the formation of polycrystal planes or geometric shapes. The films show good adhesion to the copper surface. The types of crystal structure formed are not fully identified, but according to the reported literature and based on SEM and electrical impedance characterization, there is evidence of cluster formation with polyhedral isolating structures in the film.

Novel furnaces were built based on oxidized copper surfaces that reached temperatures above 320 ${}^{{}^{\circ}}$C, with good stability and reproducibility in the heating curves and stability and durability over time. Experimental data evidence settling times of 636.8 s for the CuITF and 757.2 for the CITF. The results showed the CuITF was 16.3% faster at reaching 50% of the final temperature than CITF. The gain was 15% higher for CuITF than CITF. The CuITF achieved a 36.3% faster response time with the same power supply as the CITF. The importance of this result is highlighted when it comes to a fine temperature control in a heating device. The CuITF achieved a 12% higher temperature with the same power supply as the CITF.

## Figures and Tables

**Figure 1 materials-16-01361-f001:**
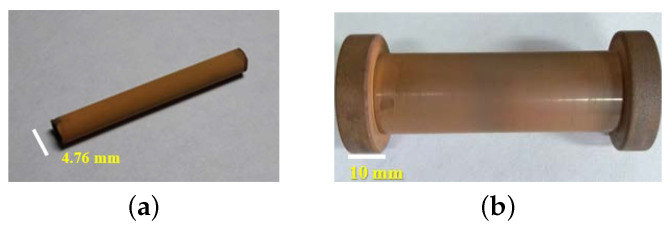
(**a**) Oxidized copper tube; and (**b**) oxidized copper cylinder.

**Figure 2 materials-16-01361-f002:**
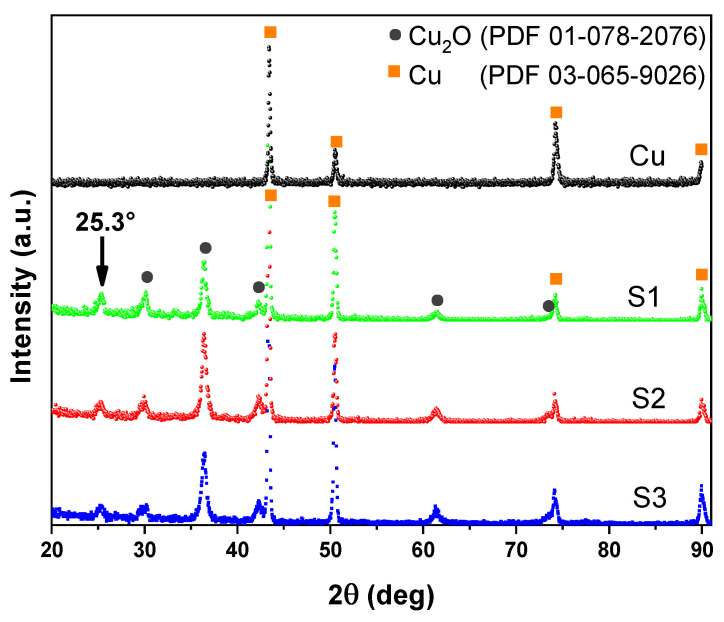
X-ray diffractogram of pure copper (Cu) and insulating oxidized copper (S1, S2, and S3).

**Figure 3 materials-16-01361-f003:**
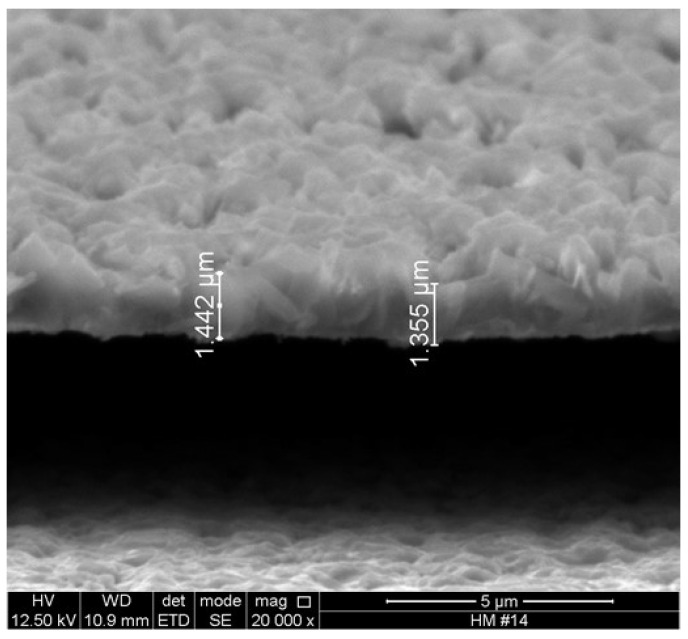
SEM image of the polyhedral Cu${}_{2}$O insulating film.

**Figure 4 materials-16-01361-f004:**
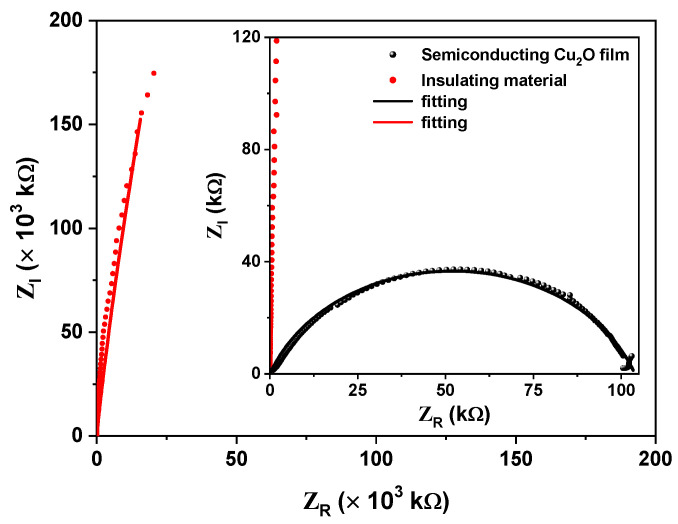
Results of electrical impedance measurements of the insulating Cu${}_{2}$O film. For comparison purposes, the inset corresponds to the fit using the circuit model on a semiconductor sample of CuO.

**Figure 5 materials-16-01361-f005:**
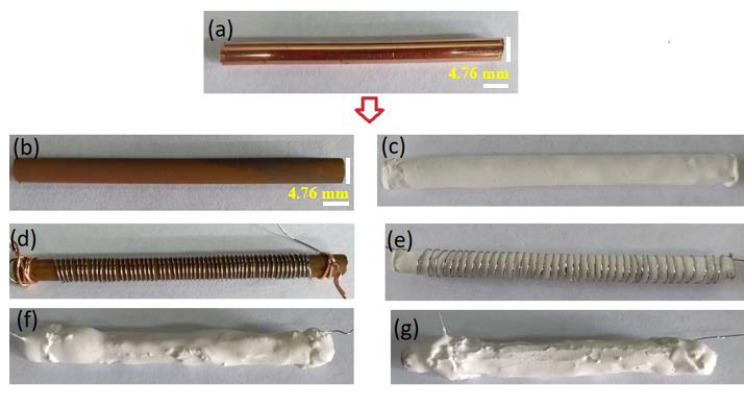
Steps to make the prototypes CuITF (left column) and CITF (right column, using ceramic coating): (**a**) the unoxidized copper tube precursor material; (**b**) the copper tube covered with an insulating copper oxide layer; (**c**) electrical insulation and ceramic thermal coupling (**d**,**e**) electric resistance heaters; (**f**,**g**) protection cover ceramic.

**Figure 6 materials-16-01361-f006:**
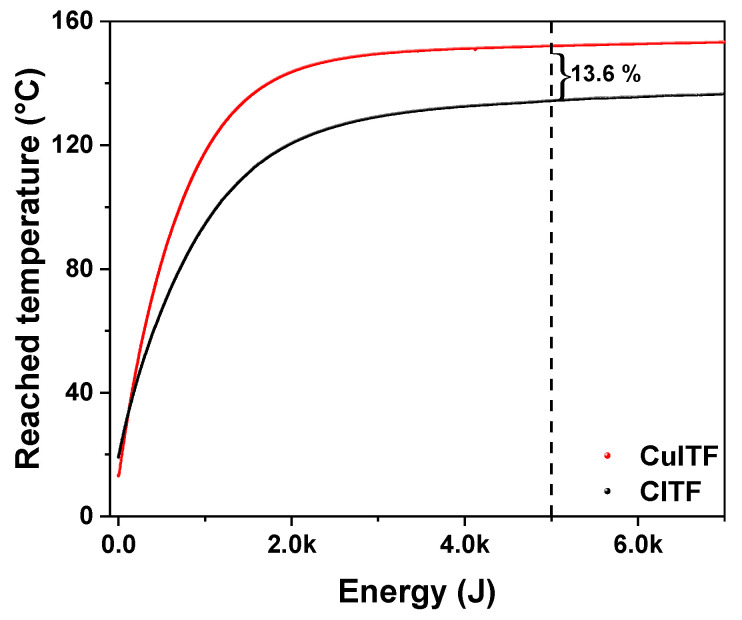
The reached temperature curves as a function of energy consumed for each furnace.

**Figure 7 materials-16-01361-f007:**
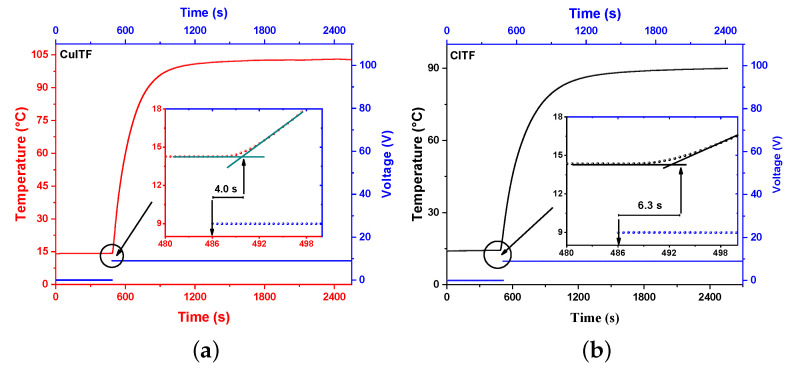
Dead time for (**a**) CuITF and (**b**) CITF.

**Figure 8 materials-16-01361-f008:**
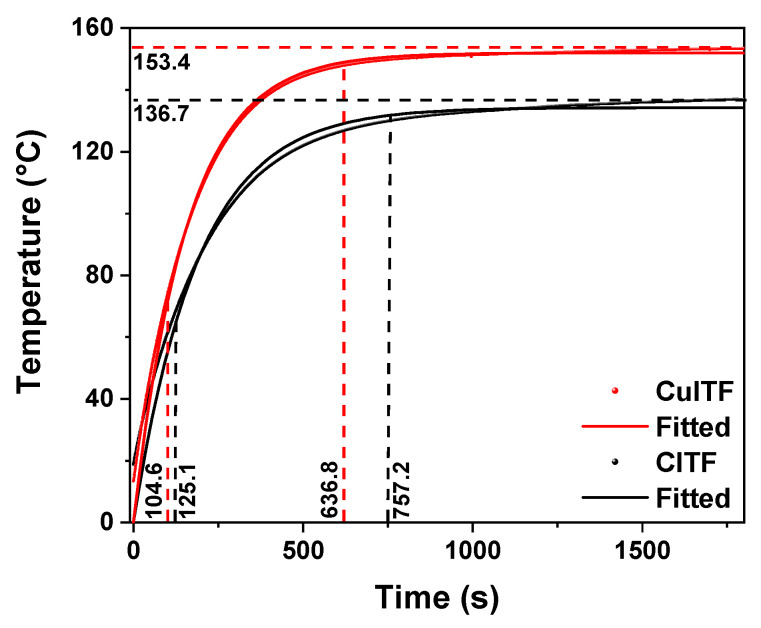
The heating curve settings for CuITF and CITF.

**Figure 9 materials-16-01361-f009:**
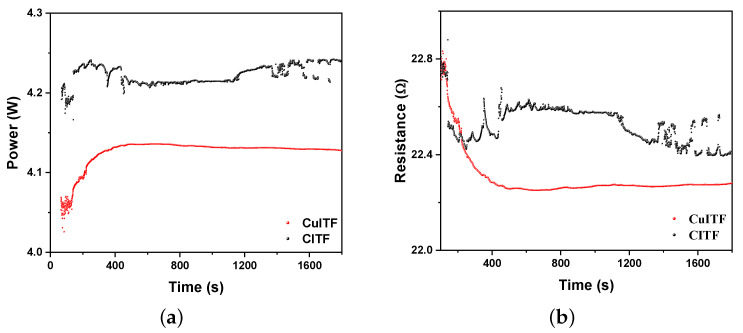
(**a**) Electrical power dissipation as a function of time in both furnaces. (**b**) Electrical resistance variations resistance as a function of time in both furnaces.

**Figure 10 materials-16-01361-f010:**
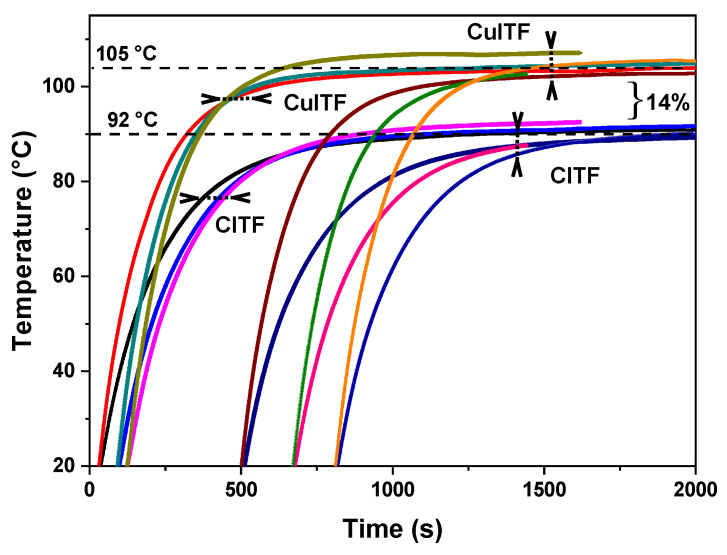
Temperature behavior as a function of time for six runs for each furnace. The displacement in time corresponds to different procedure start times.

**Figure 11 materials-16-01361-f011:**
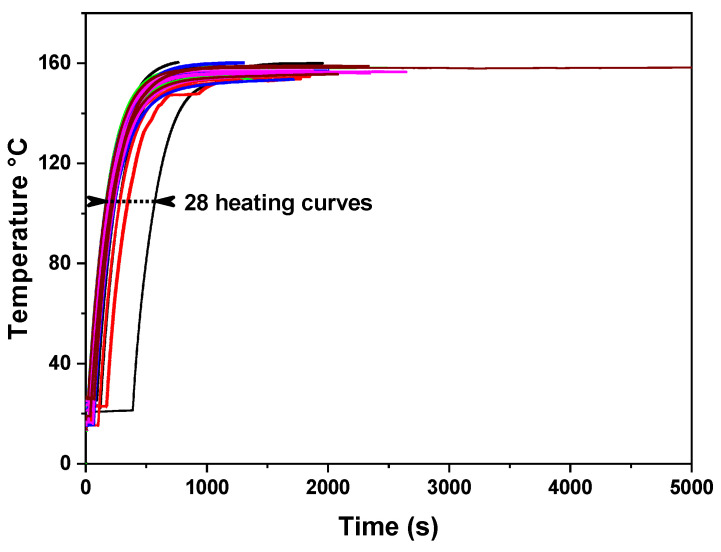
The heating curves show reproducibility and durability in time over a 2-month interval.

**Figure 12 materials-16-01361-f012:**
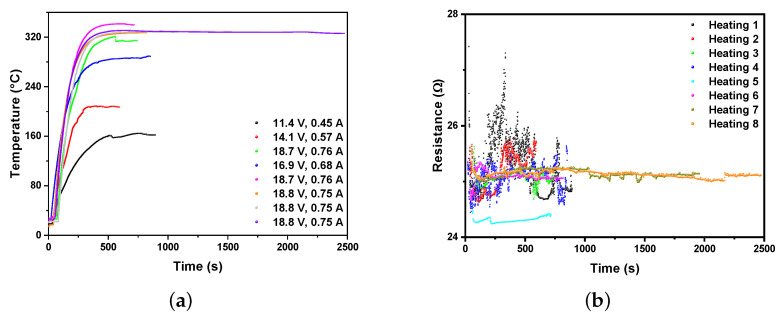
(**a**) CuITF heating curves as a function of time for different values of current and voltage, using an electrical resistance of 25 Ω; and (**b**) variation of electrical resistance as a function of time for 8 heating runs.

**Table 1 materials-16-01361-t001:** Fitting modeling parameters for the heating curve of the CuITF and CITF.

	CuITF	CITF
$T_{f}$ (${}^{{}^{\circ}}$C)	151.9	134.2
τ (s)	158.1	190.1
$R^{2}$	0.99463	0.98041

**Table 2 materials-16-01361-t002:** Main characteristics of the furnaces.

Characteristic	CuITF	CITF
Time constant, τ (s)	158.1	190.1
Delay time $t_{d}$ (s)	104.6	125.1
Settling time $t_{s}$ (s)	636.8	757.2
Dead time, $\tau_{d}$ (s)	4.0	6.3
Gain	16.0	14.2

## Data Availability

Not applicable here.

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
