# Peer review of "Procedure to Obtain Cu_2_O Isolate Films, Structural, Electrical, and Morphological Characterization, and Its Use as an Electrical Isolator to Build a New Tube Furnace"

_materials, 2023, doi:10.3390/ma16041361_

Round 1

Reviewer 1 Report

This work reported the preparation of copper oxide film on copper surface by thermal oxidation and its application in the furnace. Some characterizations were carried out. In this sense, this manuscript looks well as a whole but needs major revisions as follows before publication.

1. The novelty of this work was not described clearly in the Introduction. Authors should revise the introduction.

2. There were no scaleplate Figure 1 and authors should add it.

3. Authors should describe what the S1, S2 and S3 mean. Authors claimed that Cu3O2 or some substrate impurity such as CuHO2 or Cu-P-O was probably included in the samples. XPS characterization should be employed to identify the valence of copper in the samples.

4. Authors also should add scale in the Figure 3. Authors claimed that the formation of Cu2O crystals with different facets was observed and bipyramidal crystal structure was seen from SEM images in Figure 3. But it cannot be seen clearly. TEM characterization should be carried out.

5. The reached temperature was deeply related with thermal conductivity of the films. The thermal conductivity of the films should be measured. The curves in Figure 6 should be discussed in detail. Why did the CuITF reach a higher temperature for 5 kJ energy?

6. Some English should be improved.

Author Response

Assigned Editor

Diego Ding

Editor

Materials

Manuscript ID materials-2081261

Title: Procedure to obtain Cu2O isolate films, structural, electrical, and morphological characterization, and its use as an electrical isolator to build a new tube furnace

Authors: H. Correa, R. Pineda S., and D. Peña Lara

Dear

Editor

First, we would like to thank you for the comments and valuable suggestions, which were very useful in improving our paper.

Please find enclosed the revised version of our manuscript, modified following the comments. All points addressed were taken into consideration as follows:

Reviewer #1:

This work reported the preparation of copper oxide film on copper surface by thermal oxidation and its application in the furnace. Some characterizations were carried out. In this sense, this manuscript looks well as a whole but needs major revisions as follows before publication.

  1. The novelty of this work was not described clearly in the Introduction. Authors should revise the introduction.

The Introduction section has been improved.

…1.2 eV gap and can be synthesized in…

Tan et al. [3] studied the electrical characteristics of copper oxides showing that the octahedral crystal is highly conductive, the cubic crystal is moderately conductive, and the rhombic dodecahedron crystal is a non-conductor.

…interfaces [3]. Roy and Wright [31]…

In this work, we report a thermal oxidation procedure to produce an electrically insulating Cu2O film grown on a copper surface.

We use the Cu2O electrical insulating film to build a novel prototype of a resistive furnace.

Some references were omitted

…crystallographic forms [2,3].

….history [2,7–11].

…applications [7,12–21]…

(Paragraphs 1—6, pages 1 and 2)

  1. There were no scaleplate Figure 1 and authors should add it.

Scaleplate was included in figure 1

(Figure 1, page 3)

  1. Authors should describe what the S1, S2 and S3 mean.

The S1, S2, and S3 meaning has been included:

S1 (sample 1, green line), S2 (sample 2, red line), and S3 (sample 3, blue line)

(Paragraph 1, page 3)

Authors claimed that Cu3O2 or some substrate impurity such as CuHO2 or Cu-P-O was probably included in the samples. XPS characterization should be employed to identify the valence of copper in the samples.

We thank the reviewer for his valuable observation.

In this work, we don’t make a deep study of impurity content in the sample. The literature reports that CuHO or Cu-P-O impurities do not intervene in the electrical conductivity, isolating the film from the impurity content [1].

The XPS technique provides information on the copper valence; however, the literature suggests using LSV (Linear scanning voltammetry) [1—3] for this purpose. We expect to do future detailed study work, including this characterization.

[1] Mencer, D. E.; Hossain, M. A.; Schennach, R.; Grady, T.; McWhinney, H.; Gomes, J. A. G.; Kesmez, M.; Parga, J. R.; Barr, T. L.; Cocke, D. L. On the surface analysis of copper oxides: the difficulty in detecting Cu3O2, Vacuum, 2004, 77, 27–35.

[2] D. L.; Schennach, R.; Hossain, M. A.; Mencer, D. E.; McWhinney, H.; Parga, J. R.; Kesmez, M.; Gomes, J. A. G.; Mollah, M. Y. A. The low-temperature thermal oxidation of copper, Cu3O2, and its influence on past and future studies, Vacuum, 2005, 79, 71–83.

[3] Cocke, D. L.; Mencer, D. E.; Hossain, M. A.; Schennach, R.; Kesmez, M.; Parga, J. R.; Naugle, D. G. Investigation of the Metal–Oxide Buried Interfacial Zone with Linear Sweep Voltammetry, J. Appl. Electrochem., 2004, 34, 919–927.

(page 3)

  1. Authors also should add scale in the Figure 3.

Scaleplate was included in figure 3.

(Figure 3, page 3)

Authors claimed that the formation of Cu2O crystals with different facets was observed and bipyramidal crystal structure was seen from SEM images in Figure 3. But it cannot be seen clearly.

We agree. We see, actually, other crystalline architectures different from bipyramids. We have observed a certain predominance to be bipyramidal in our films, but this observation is not conclusive. The experimental evidence shows that the films with these architectures are electrical insulators. In our work text,  we are going to replace bipyramidal word with polyhedral.

New SEM images have been included for your clarity.

TEM characterization should be carried out.

We didn’t plan this study because the films are joined to the Cu substrate

  1. The reached temperature was deeply related with thermal conductivity of the films. The thermal conductivity of the films should be measured.

In the literature, the thermal conductivity of Cu2O is reported. It Is higher than ceramic or other isolated compounds used in these applications.

The curves in Figure 6 should be discussed in detail. Why did the CuITF reach a higher temperature for 5 kJ energy?

The first paragraph on page 5 has a mistake. This paragraph has been improved:

The curves show that, with a similar electrical energy supply for both furnaces, the CuITF reaches an average temperature 13.6% higher than the CITF for a certain representative energy value of 5 kJ in the range of temperature stabilization.

(Paragraph 1, page 5)

  1. Some English should be improved.

The English was improved.

Reviewer 2 Report

The manuscript reports the procedure for the formation of Cu2O films to be used in a new type of tube furnace.

The topic of manuscript is within a scope of the journal and it can be interesting to a broad readership.

However, several important issues should be adressed before the manuscript can be accepted for publication in Materials.

1. The introduction section should be rewritten. In the present form it is a simple collection of paragraphs describing several topics. Although these topics are connected with the subject of manuscript, the main scientific idea of the paper is almost lacking in the present form. Moreover, this part can be significantly shortened.

2. Figure 2 - please add PDF cards numbers for both Cu and Cu2O to the graph.

3. Since as it is mentioned in the introduction section, the conductivity of Cu2O depends also on the crystallite orientation preference (see for instance https://doi.org/10.1021/acs.nanolett.5b00150). This issue should be also considered in discussion

4. In general, the material characterization should be extended since in the present form the manuscript is rather a technical report instead of scientific paper.

5. What about the reproducibility of the thermal oxidation procedure?

6. Conclusions seem to be extended abstract, so this section should be also rewritten to show the most significant findings of the paper.

7. Do the Authors consider some other methods of generation of copper oxide films on Cu instead of thermal treatment?

Author Response

Assigned Editor

Diego Ding

Editor

Materials

Manuscript ID materials-2081261

Title: Procedure to obtain Cu2O isolate films, structural, electrical, and morphological characterization, and its use as an electrical isolator to build a new tube furnace

Authors: H. Correa, R. Pineda S., and D. Peña Lara

Dear

Editor

First, we would like to thank you for the comments and valuable suggestions, which were very useful in improving our paper.

Please find enclosed the revised version of our manuscript, modified following the comments. All points addressed were taken into consideration as follows:

Reviewer #2:

This work reported the preparation of copper oxide film on copper surface by thermal oxidation and its application in the furnace. Some characterizations were carried out. In this sense, this manuscript looks well as a whole but needs major revisions as follows before publication.

  1. The introduction section should be rewritten. In the present form it is a simple collection of paragraphs describing several topics. Although these topics are connected with the subject of manuscript, the main scientific idea of the paper is almost lacking in the present form. Moreover, this part can be significantly shortened.

The Introduction section has been improved.

…1.2 eV gap and can be synthesized in…

Tan et al. [3] studied the electrical characteristics of copper oxides showing that the octahedral crystal is highly conductive, the cubic crystal is moderately conductive, and the rhombic dodecahedron crystal is a non-conductor.

…interfaces [3]. Roy and Wright [31]…

In this work, we report a thermal oxidation procedure to produce an electrically insulating Cu2O film grown on a copper surface.

We use the Cu2O electrical insulating film to build a novel prototype of a resistive furnace.

Some references were omitted 

…crystallographic forms [2,3].

….history [2,7–11].

…applications [7,12–21]…

(Paragraphs 1—6, pages 1 and 2)

  1. Figure 2 - please add PDF cards numbers for both Cu and Cu2O to the graph.

PDF cards were included.

(Figure 2, page 3)

  1. Since as it is mentioned in the introduction section, the conductivity of Cu2O depends also on the crystallite orientation preference (see for instance https://doi.org/10.1021/acs.nanolett.5b00150). This issue should be also considered in discussion

The literature shows evidence of a relationship between the facet architecture and the electrical conductivity properties of Cu2O crystals [7]. We have observed that a mixture of architectures with a predominant bipyramidal is an electrically insulating material, but this observation is inconclusive. We found other crystal forms different from the bipyramidal architectures.

The literature shows evidence of a relationship between the facet architecture and electrical conductivity properties of Cu2O crystals [3]. In our isolator films, we observed a mixture of polyhedral architectures; however, this observation is not conclusive to argue how is the relationship between the architecture of our samples with conductivity.

For your clarity we include a supplemental updated SEM micrograph.

 (Paragraph 1, page 4)

  1. In general, the material characterization should be extended since in the present form the manuscript is rather a technical report instead of scientific paper.

We agree.

We hope to make a future detailed study work, including the material characterization.

The paper seems to be a technical report due we include a technology application. We hope to make a future detailed study work, including the material characterization.

  1. What about the reproducibility of the thermal oxidation procedure?

The thermal oxidation procedure is reproducible.

In the final paragraph of section result, page 2, we mentioned:

... the thermal oxidation method, obtaining reproducibility of the oxide film... over time.

  1. Conclusions seem to be extended abstract, so this section should be also rewritten to show the most significant findings of the paper.

Thanks for your valuable suggestions

The Conclusions section has been improved.

We reported a novel procedure for the synthesis by thermal treatment of a Cu2O electrical insulator film on copper surfaces using appropriate parameters such as heating profiles and oxidation temperature in the air atmosphere. This film has an electrical resistance of the order of the mega-ohm suitable to be used as electrical insulation in the manufacture of furnaces.

The results of the structural characterization by X-ray diffraction of the oxidized copper films show the presence of cuprous oxide (Cu2O). The procedure to obtain the insulating copper oxide film does not require sophisticated technological equipment since the process is carried out in an atmosphere of air and heating at relatively low temperatures. It is an economical and easy-to-implement technique.

Novel furnaces were…

(Conclusions section, page 10)

  1. Do the Authors consider some other methods of generation of copper oxide films on Cu instead of thermal treatment?

There are other methods as electrodeposition and sputtering, but for technical reasons and costs, we consider that thermal oxidation presents the following advantages: Rapid large-scale production of metal oxide, the simplest routes to synthesize, this technique involves the direct heating with low energy cost, metal oxide nanostructures production without using hazardous chemicals.

Reviewer 3 Report

It is a good and clear work, is structured and quite complete. However, there is not any discussion or comparation these results with other synthesis methodologies, other materials, or other works. This method of synthesis has been tested in others works.

By other hand, I'd like that authors’ have taken in account concepts as efficiency or sustainability on chemistry, for example how this synthesis method is more sustainability? 

Finally, references must be improved, select the most relevant for the aims of the paper.

Author Response

Assigned Editor

Diego Ding

Editor

Materials

Manuscript ID materials-2081261

Title: Procedure to obtain Cu2O isolate films, structural, electrical, and morphological characterization, and its use as an electrical isolator to build a new tube furnace

Authors: H. Correa, R. Pineda S., and D. Peña Lara

Dear

Editor

First, we would like to thank you for the comments and valuable suggestions, which were very useful in improving our paper.

Please find enclosed the revised version of our manuscript, modified following the comments. All points addressed were taken into consideration as follows:

Reviewer #3:

It is a good and clear work, is structured and quite complete.

Response to general observations reviewer

It is a good and clear work, is structured and quite complete. However, there is not any discussion or comparation these results with other synthesis methodologies, other materials, or other works. This method of synthesis has been tested in others works. By other hand, I'd like that authors’ have taken in account concepts as efficiency or sustainability on chemistry, for example how this synthesis method is more sustainability? Finally, references must be improved, select the most relevant for the aims of the paper.

There are other methods as Electrodeposition and Sputtering, but for technical reasons and costs, we consider that thermal oxidation presents the following advantages: Rapid large-scale production of metal oxide, the simplest routes to synthesize, this technique involves the direct heating with low energy cost, metal oxide nanostructures production without using hazardous chemicals 

Although other authors have used the thermal synthesis method, it has not been used for the purpose of this work. (growing electrical insulating film and furnace building).

Some references were omitted:

References from 3 – 6, 13 – 1, 19, 24 – 27, and 30 were omitted.

(Reference section, pages 10—12)

Responses to the pdf observations

There is not any discussion or comparation these results with other synthesis methodologies, other materials, or other works. This method of synthesis has been tested in others works.

By other hand, I'd like that authors’ have taken in account concepts as efficiency or sustainability on chemistry, for example how this synthesis method is more sustainability?

There are other methods as Electrodeposition and Sputtering, but for technical reasons and costs, we consider that thermal oxidation presents the following advantages: Rapid large-scale production of metal oxide, the simplest routes to synthesize, this technique involves the direct heating with low energy cost, metal oxide nanostructures production without using hazardous chemicals

Although other authors have used the thermal synthesis method, it has not been used for the purpose of this work.

On text you said "The CuITF presents higher temperature gain than the CITF". However on fig 10,  CITF results showed 3 temperatures around 100º and other 3 around 92, the same that CuUTF. In my opinion figure 10 doesn't support your affirmation, maybe you could getting better it or adding more information with real values.

We agree, the mistake in figure 10 was corrected

If authors want to compare electrical different conditions i suggest use always the same experimental time. It doesn't have sense that's just one experiment was carried until 25000s.

The stability of resistance in all temperature ranges can be observed in figure 12 b. We do not include this information in figure 12 a by not overcrowding the figure

The curves that reach the 2500 s were the last to be measured. So, in the rest of the curves, the resistance must be constant.

Same thing. why are there only two samples until 2500s. Authors shall homogenized or justify this experimental variations.

The same as in the previous question

Which references?  Any way, in the epigraph of conclusion references must be avoid

Thanks by your comment. The references will be omitted in results

Reviewer 4 Report

The manuscript (materials-2081261) under title “Procedure to obtain Cu2O isolate films, structural, electrical, and morphological characterization, and its use as an electrical isolator to build a new tube furnace”. The following are my comments and critique:

Abstract is good.

Introduction section is good.

Material and methods are good.

Results and discussion:

The characterization section is good.

Figure 5, please explain what the meaning of letters in caption figure.

Conclusion is good.

Tables and Figures are good.

Author Response

CapAssigned Editor

Diego Ding

Editor

Materials

Manuscript ID materials-2081261

Title: Procedure to obtain Cu2O isolate films, structural, electrical, and morphological characterization, and its use as an electrical isolator to build a new tube furnace

Authors: H. Correa, R. Pineda S., and D. Peña Lara

Dear

Editor

First, we would like to thank you for the comments and valuable suggestions, which were very useful in improving our paper.

Please find enclosed the revised version of our manuscript, modified following the comments. All points addressed were taken into consideration as follows:

Reviewer #4:

The manuscript (materials-2081261) under title “Procedure to obtain Cu2O isolate films, structural, electrical, and morphological characterization, and its use as an electrical isolator to build a new tube furnace”. The following are my comments and critique:

  1. Abstract is good.

  1. Introduction section is good.

  1. Material and methods are good.

  1. Results and discussion:

  1. The characterization section is good.

  1. Figure 5, please explain what the meaning of letters in caption figure.

Caption of figure 5 was completed

Figure 5 Steps to make the prototypes CuITF (left column) and CITF (right column, using ceramic coating).a) The unoxidized copper tube precursor material, b) The copper tube covered with an insulating copper oxide layer, c) Electrical insulation and ceramic thermal coupling d -e) Electric resistance heaters, f -g) protection cover ceramic

  1. Conclusion is good.

  1. Tables and Figures are good.

Reviewer 5 Report

' oxidation to grow the copper (I) oxide phase o...' what was the number in the abstract indicate?

You can improve the section of abstract by adding some of main results.

I have noticed the section of materials and methods comes after results and discussion section!!!  Are the results achieved before experiment have done!!!! you can remove the section of Materials before results and discussion.

Conclusion can be summarized well more than exist one 

Author Response

Assigned Editor

Diego Ding

Editor

Materials

Manuscript ID materials-2081261

Title: Procedure to obtain Cu2O isolate films, structural, electrical, and morphological characterization, and its use as an electrical isolator to build a new tube furnace

Authors: H. Correa, R. Pineda S., and D. Peña Lara

Dear

Editor

First, we would like to thank you for the comments and valuable suggestions, which were very useful in improving our paper.

Please find enclosed the revised version of our manuscript, modified following the comments. All points addressed were taken into consideration as follows:

Reviewer #5:

  1. oxidation to grow the copper (I) oxide phase o...' what was the number in the abstract indicate?

The notation has been rewritten.

…the Cu2O on copper…

  1. You can improve the section of abstract by adding some of main results.

The abstract has been modified; the principal results have been highlighted.

  1. I have noticed the section of materials and methods comes after results and discussion section!!! Are the results achieved before experiment have done!!!! you can remove the section of Materials before results and discussion.

It is journal policy to have the materials and methods section follow the results and discussion section.

  1. Conclusion can be summarized well more than exist one

The Conclusions section has been improved.

We reported a novel procedure for the synthesis by thermal treatment of a Cu2O electrical insulator film on copper surfaces using appropriate parameters such as heating profiles and oxidation temperature in the air atmosphere. This film has an electrical resistance of the order of the mega-ohm suitable to be used as electrical insulation in the manufacture of furnaces.

The results of the structural characterization by X-ray diffraction of the oxidized copper films show the presence of cuprous oxide (Cu2O). The procedure to obtain the insulating copper oxide film does not require sophisticated technological equipment since the process is carried out in an atmosphere of air and heating at relatively low temperatures. It is an economical and easy-to-implement technique.

Novel furnaces were…

(Conclusions section, page 10)

Round 2

Reviewer 2 Report

Now, the mansucript is acceptable

Author Response

There are no comments from reviewer 2; however, we attach scale bars in Fig. 5 in the final version of our manuscript.
